# Barriers to Couplet Care of the Infant Requiring Additional Care: Integrative Review

**DOI:** 10.3390/healthcare11050737

**Published:** 2023-03-02

**Authors:** Amanda Curley, Linda K. Jones, Lynette Staff

**Affiliations:** School of Nursing, University of Tasmania, Launceston, TAS 7248, Australia

**Keywords:** couplet care, mother–baby care, mother-infant dyad, family-centred care, single room maternity care model, family-integrated care, rooming in

## Abstract

Background: Historically, once the baby was born, the mother and baby were separated shortly after birth into a postnatal ward and a baby nursery. Overtime, with advances in neonatology led to an increasing number of neonates being separated from their mothers at birth for specialised neonatal care if they required additional needs. As more research has been undertaken there is an increasing focus that mothers and babies should be kept together from birth, termed couplet care. Couplet care refers to keeping the mother and baby together. Despite this evidence, in practice, this is not happening. Aim: to examine the barriers to nurses and midwives providing couplet care of the infant requiring additional needs in postnatal and nursery. Methods: A thorough literature review relies on a well-developed search strategy. This resulted in a total of 20 papers that were included in this review. Results: This review revealed five main themes or barriers to nurses and midwives providing couplet care: models of care, systems and other barriers, safety, resistance, and education. Discussion: Resistance to couplet care was discussed as being caused by feelings of lack of confidence and competence, concerns around maternal and infant safety and an under-recognition of the benefits of couplet care. Conclusion: The conclusion is that there is still a paucity of research in relation to nursing and midwifery barriers to couplet care. Although this review discusses barriers to couplet care, more specific original research on what nurses and midwives themselves perceive to be the barriers to couplet care in Australia is needed. The recommendation is therefore to undertake research into this area and interview nurses and midwives to ascertain their perspectives.

## 1. Introduction

Couplet care is defined as care that is provided to both mother and baby after birth in the same room by the same midwife without separation [1,2,3]. The benefits of keeping mothers and their babies together after birth are well recognised [1]. Although there is an abundance of evidence on the benefits to both mother and baby, a disconnect remains, however, between what the evidence states and what occurs in practice. Midwives and nurses are perfectly positioned to advocate for and influence the uptake of the mother-baby, or couplet care model, to be implemented in practice however, it is not consistently provided [4,5,6]. It is therefore important to understand the barriers that prevent the evidence-based practice in the clinical environment, with the hope that identification of those barriers will draw attention to this important issue and affect change in Australian maternity units. This integrative review, enabling a broad review of the literature and facilitating a comprehensive understanding [7], will examine the literature on the barriers to midwives and nurses providing couplet care where infants have been identified as having additional care needs. The aim is to narrate those findings in a thematic way to give an overview of the state of the evidence today and discuss the barriers to couplet care. The literature is generally international with some specifically related to Australia as this forms part of an honours dissertation undertaken in Australia that interviewed midwives and nurses to identify barriers to couplet care from their perspective. By nature, this paper is very Western focused, as it is not within the scope of the research to go beyond this.

## 2. Background

Postnatal care of the mother and baby has evolved over time. The pre-medical model practiced involved antenatal, birth, and postnatal care occurring in the home. This was the case until a few decades into the 20th century, when care began moving into the hospital environment [8,9]. In the 1940s, the first mother–baby units were described in mental health hospitals, and the first steps towards mother–baby care were taken in the United States, which came to be known as ‘rooming-in’ [9,10,11]. Although rooming-in gained attention for being a more holistic model of care, new mothers were often left to take care of their new baby without any guidance or support from the midwifery staff [9]. Furthermore, care was provided to mothers and babies in separate locations, leading to missed educational opportunities and the development of mother-crafting skills [9,10]. Around the 1970s the concept of family-centred care started to develop which saw different models emerge, such as the single-room maternity care model [1,9,11]. Single-room maternity care was a model that provided care to the mother and baby in the same room [11]. This facilitated bonding and attachment, breastfeeding, and encouraged education; however, care remained fragmented [9]. In this model, care was provided by individual specialty teams; the postpartum team provided care to the women, and the nursery staff provided care to the baby [9]. This model perpetuated the disconnect in care and continued to treat the mother and baby separately from each other, not as the intertwined unit that they are now known to be.

Family-centred care continued to gain attention over recent years, and with the rise in feminism came the increasing demands for a philosophical change from traditional healthcare worker-centred, hospital-centred care to care that focussed on the mother-baby couplet in the context of the whole family unit [9,12]. In the late 1990s, Phillips [13] further defined mother–baby care under the tenets of the family-centred care philosophy, which emphasised safe, quality, mother-baby care that promoted family unity [9]. It was this philosophical change that instigated the evolution of family-centred and now mother–baby-centred care, or the more recently coined term ‘couplet care’ [13]. 

The benefits of couplet care include that it supports bonding and attachment, early flora colonisation for immune protection, breastfeeding through the hospital stay and after discharge, increased women’s satisfaction, improved care coordination, reduced communication gaps, promoted teaching opportunities for the women, increased women’s confidence in caring for their infants, and increased staff efficiencies with reduced costs [1].

Some of the disadvantages are the opposite of the benefits and include a disconnect in care, treating mother and infant separately, missed educational opportunities and development of mother-crafting, interrupts attachment and breastfeeding, exposure of the infant to other flora and risk of infection, maternal anxiety increase, decreased productivity and increased staff costs [1].

As recognition of the benefits of couplet care grows, so does its popularity and use in maternity units worldwide [3]. Despite this, there remains a large cohort of mothers and babies that are routinely separated without compelling medical reasons or separated based on a set of criteria determined by individual healthcare facilities [14,15]. For example, a baby may be born less than a particular weight or earlier than a set gestation without taking into consideration individualised care and the needs of the mother–baby dyad [14,16,17]. There have been some attempts in maternity units to provide care for infants who require phototherapy or antibiotics in the postnatal ward with the mother. Often, however, there is still the tendency for the nurse from the nursery to come to the ward to oversee the infants’ care [5].

With the rate of late preterm births rising and demands on postnatal and neonatal beds increasing, a closer examination of the systemic benefits as well as the mother and baby benefits of couplet care is warranted. According to the Australian Institute of Health and Welfare [18], 18,546 late preterm infants were born between 34 weeks and 0 days, and 36 weeks and 6 days in Australia in 2018. The Australian Institute of Health and Welfare [18] also states that preterm babies have a 72% chance of being admitted to the neonatal nursery. Admitting a baby to the neonatal nursery interrupts mother–infant attachment, which can have lifelong impacts on both mother and infant [19]. Evidence suggests that infants and mothers who were separated during their hospital stay had decreased opportunities for skin-to-skin contact and kangaroo mother care (KMC), resulting in decreased breastfeeding rates and breastfeeding success [20]. Infants identified as having additional care needs such as late preterm, small for gestational age and infants who could benefit from early flora colonisation for immune protection from skin-to-skin contact, are particularly affected by separation from their mother [20]. Infants deprived of this contact with their mother demonstrated increased incidences of hypothermia, hypoglycaemia, and hyperbilirubinemia [20]. Maternal effects of separation included increased maternal anxiety, postnatal depression, decreased maternal confidence, decreased breast milk supply and increased rates of mastitis [20]. 

The benefits of keeping mother and baby together are well understood and recognised, yet the practice is not consistent, especially for babies requiring additional care such as the late preterm infant [21]. There are currently only minimal maternity units that provide couplet care [8]. Considering the benefits, this review intends to examine the barriers to nurses and midwives providing couplet care for infants requiring additional needs. The following section will describe the search strategy used to gather the literature to inform this review.

## 3. Materials and Methods

An integrative review methodology was utilised for this paper as it enables a broad review of the literature through a combination of methodologies and therefore facilitates a comprehensive understanding of barriers to couplet care [7]. The literature was gathered using Arksey and O’Malley’s [22] 5-step framework, and Levac et al.’s [23] method of synthesising health evidence. Generally, international literature was sourced with a specific focus on Australia, as this was the setting for the next stage of the research. A thorough literature review relies on a well-developed search strategy. For this literature review, a PEO was developed to provide a framework for the search strategy (Table 1.). This was preferred due to the focus being more on qualitative research [24]. The target population for this review is the midwife working in postnatal wards, and nurses working in the nursery, and their problem was caring for mothers and their babies who have additional care needs (see definition). The exposure was to couplet care, and the outcome was to identify the barriers to couplet care.

A thorough literature search was conducted using the databases CINAHL, PubMed, Medline, EBSCOhost and the Cochrane Library for systematic reviews. Scopus was also used following the evaluation of the articles gathered from the primary search. Reference lists were scanned for relevant material. Grey literature was included for completeness such as Google Scholar, government websites and professional and regulatory bodies. A search of the databases listed in Table 2 included main subject terms as well as MeSH terms, depending on the database. Nurs AND/OR midwife AND postnat OR postpartum AND couplet care OR couplet OR rooming-in OR mother-baby care OR mother-infant dyad AND barriers AND/OR facilitator OR enablers AND late preterm OR preterm AND bab OR Infant OR Neonat All styles of literature were included with no restrictions placed on its origin; however, the literature that was not available in English was excluded. Truncation, boolean operators, and wild cards were also used to maximise results. General limiters included articles available in the English language and peer-reviewed articles. A date range limiter was applied originally, however, was removed during the search to enable the gathering of as much empirical data as possible. 

### 3.1. Screening

This primary search yielded 252 papers across the databases to review. Duplicate articles were removed to reveal 230 papers. The titles of these articles were then scanned for relevance, leaving 101 articles for further examination. Then abstracts of these articles were scanned and reviewed against the inclusion criteria, removing a further 24 papers, leaving 77 papers for further review. The remaining papers from this process had a full-text review and were assessed for their suitability for inclusion. Following these assessments, 18 papers were removed due to contextual irrelevance such as the location of care. Lastly, 47 papers were removed because, although they did refer to maternity care, there was no reference to couplet care or mother–baby care. This left 12 papers for inclusion. The reference lists and citations of these papers were checked for any further papers suitable for inclusion. Through this scanning, 8 further papers were found, resulting in a total of 20 papers that were included in this review. 

### 3.2. Inclusion and Exclusions 

For this review, parameters around the infant population were required to ensure relevant literature was retrieved. Included in this review are infants defined as late preterm from 34 weeks and 0 days gestation until 36 weeks and 6 days [25]. Clinical conditions were also considered in this cohort and included hypoglycaemia, hypothermia, hyperbilirubinemia, and mild respiratory distress. Publications were excluded if they included infants requiring intensive care, i.e., ventilatory support, birth weight less than 2000 g, and gestational age at birth less than 34 weeks and 0 days. Infants born with congenital abnormalities or categorised as needing high levels of care such as those diagnosed with severe hypoxic injury at birth were also excluded. The PRISMA flow diagram in Figure 1 shows an infographic of this selection process.

Summary of articles appear in the following table.

## 4. Results

The final literature selected for this review included a wide range of theoretical and empirical work (see Table in the Appendix). This included a systematic review [11,26,27], a pilot study [21], quantitative of units [16], a mixed method of units [28,29], interviews [20,30], pre-post-intervention on knowledge [31,32], retrospective chart review [14,15,17,33], scoping review [34,35], and commentary [5,9,36]. Key points from these articles were then themed. This review revealed five main themes: models of care, systems and other barriers, safety, resistance, and education.

### 4.1. Models of Care 

When examining couplet care of late preterm and term infants, it is important to understand the concept of models of care and to understand the environment in which they occur. Quality maternity care ensures that women, babies and families receive care and support that considers their values, cultures, desires, circumstances and clinical needs [35]. A maternity model of care is a concept that encompasses care during the antenatal, perinatal and postnatal periods of a woman’s childbearing [35]. Historically, the focus of these maternity models of care has been on antenatal care and childbirth with limited focus on the postnatal period with a wide range of variables across care provision [28]. Recently however, postnatal models of care have been gaining more attention as maternity and neonatal services think of creative ways to provide evidence-based care to mothers and babies and reduce the demand for maternity and neonatal beds [1,9,11,29,33,34,37]. 

The concept of transitional beds and integrated maternity and neonatal units is emerging as innovative solutions to the postnatal care provision of mother–baby couplets as well as responding to pressure on maternity services. A transitional bed is a temporary bed that provides increased monitoring and observation for infants that may not be initially eligible for couplet care but also do not require an admission to the neonatal nursery [33]. Baker, Parker and Alissa [33] discussed the use of transitional beds as a means of avoiding expensive, low-value care and an admission to the nursery for infants with a prolonged transition to extrauterine life. This innovative approach enabled a total of 194 infants born in the year following the implementation of their study to utilise the transitional bed and 144 of those were returned to couplet care with their mothers [33]. One of the limitations of this study was that, although it alluded to returning to couplet care, it was not specific as to how long infants were in transitional beds before being returned to their mothers, only stating that those who required respiratory support beyond 4 h would require admission to the neonatal nursery [33]. 

de Rooy and Johns [34] also discussed transitional care; however, the model they mentioned had different staff caring for either the mother or the baby. In this case, a midwife contributing expert knowledge of maternal care and a neonatal nurse contributing their skills and knowledge in looking after preterm or vulnerable infants worked together to care for the mother and baby. This could be suggestive of inflexibility in the workforce to provide couplet care but rather single-room maternity care. Hubbard [17], however, had discussed that after two years of operation, the transitional care unit (TCU) and transitional care beds they implemented to aid in reducing unnecessary admissions to the neonatal unit were found to be unsustainable due to staffing problems and resources. They found that having the original unit under the governance of the nursery rather than the maternity unit eventuated in the beds being used as a step down from higher neonatal care rather than used as a transitional bed for babies to be returned to their mothers once babies had stabilised [17]. 

### 4.2. Systems and Other Barriers 

There are a number of other reoccurring themes in the literature regarding barriers to couplet care. Healthcare systems face barriers such as inadequate or absent guidelines, policies and procedures and limited investment in training and education to build a skilled workforce and an under-recognition of the importance and benefits of couplet care, all featured prominently in the literature [38]. On the other hand, nursery admissions yield a high economic burden and have been shown to increase the risk of infection, generate iatrogenic costs such as those associated with medication errors and increase the length of stay [14]. Additionally, the impacts of separation on the mother–infant dyad must not be underestimated. The interruption of maternal–infant attachment and bonding as well as the interruption to breastfeeding have both immediate and long-term consequences [14,15,39]. 

Criteria-led admission of infants has become a useful tool in triaging infants for admission to the nursery or returning to couplet care with their mother in some healthcare facilities [14,15,17]. Further studies, however, demonstrate that this concept does not take into consideration infants who fit into a set of criteria. For example, the late preterm baby or infants at risk of neonatal abstinence syndrome who could return to couplet care with their mother [16,17,33]. In contrast, Haidari et al. [14] found in their research a trend towards increasing admissions to neonatal units of infants with higher gestational age, heavier birth weights and lower acuities, provoking concern around the possible misuse of neonatal beds [14]. Similarly, Ziegler et al. [15] also found that there were a significant number of infants admitted to the neonatal unit without an identifiable cause. Their study showed a significant variation in admission rates that cannot be explained by definitive and compelling clinical indicators [15]. Interestingly, however, Fleming et al. [16] conducted a national survey of the admission practices for late preterm infants in England and found that one of the biggest issues clinicians face regarding this late preterm population is deciding which infants require admission to a neonatal special care nursery and those who can be safely cared for on the postnatal wards. This could be more an issue of the staff’s comfort zone and confidence in caring for these infants than anything else. Medical officers, midwives and nurses are often faced with the challenge of triaging infants against other complexities of the healthcare system, including patient safety and clinical need, efficiency, equity, patient-centred care, policy, funding, and cost [28]. This is further compounded by pre-existing issues, including the availability of skilled staff and the absence of legislated ratios [28,40].

### 4.3. Safety 

The provision of safe, quality care is fundamental to nursing and midwifery practice [41,42]. It is therefore unsurprising that safety was identified as a prominent barrier to couplet care in this review. Maternal well-being and mother and infant safety were major concerns identified. Fatigue, sleep deprivation and bed sharing, particularly during breastfeeding, were discussed as potential risks to infant safety and maternal psychological harm [5,43,44,45]. In an opinion piece written by Dalton and Maloney [5], they claimed that skin-to-skin contact and breastfeeding without adopting safety mechanisms places infants at significant risk in postnatal wards in the mother–baby care models. Since the introduction of the World Health Organisation’s (WHO) baby-friendly health initiative (BFHI), there has been a focus on the non-separation of mothers and babies to facilitate, promote, support and protect breastfeeding [46]. Prolonged skin-to-skin contact and bed sharing are encouraged under the principles of the BFHI, which is revered for its health benefits and improved mother–baby outcomes [44,46,47]. It is, however, not without its criticisms. A study undertaken by Thach [45] investigated the cause of 15 infant deaths in postnatal wards in BFHI-accredited facilities in America and found that of the 15 deaths, it was determined that death by accidental suffocation was likely to be the cause. The author had strong recommendations around the implementation of measures that increase infant safety as a result. Dalton and Maloney [5] further expressed that the couplet care model offered little to no flexibility to account for the individual needs of mothers and their babies. In effect, the principles of family-centred care, which were cited as the justification for couplet care, were unable to be met [5]. 

### 4.4. Resistance 

Resistance to change generally in healthcare remains a difficult obstacle to overcome [48]. This is despite the nursing and midwifery ethos to provide patients with contemporary, safe, high-quality evidence-based care [41,42]. A series of articles highlighted that resistance remains a key barrier to the implementation of couplet care [3,6,9,17,26,30]. Resistance to mother–baby care, in particular KMC, was discussed in Chan et al. [26], which cited healthcare workers’ under-recognition of the benefits of mother–baby care as the main cause of resistance. The author stressed that KMC was perceived by healthcare workers as substandard and that this model of care would disadvantage mothers if they had to remain in hospital beyond their own discharge [26]. 

It was revealed in another article that staff resistance to couplet care was more about the midwives’ feelings related to confidence and competence in caring for unwell infants in the postnatal ward and that their depth of knowledge of this cohort was mostly superficial [32]. Although not explicitly discussed in the article, it could be hypothesised that resistance in this instance was the product of educational flaws rather than simply resisting to provide couplet care or accept change. 

### 4.5. Education

Even though education was identified as a component of resistance related to confidence and competence, it came through the literature as a strong barrier to couplet care. Investment in nursing and midwifery education is intrinsically fundamental to ensuring nurses and midwives continue to provide high-quality culturally sensitive, safe, and respectful maternity care [36]. With the support of the International Confederation of Midwives (ICM) [49] and the WHO [46], there is global engagement to further strengthen the midwifery workforce and continue the expansion of the profession in numbers, capacity, and competence. Despite this, there remain inconsistencies in the education of midwives globally [50]. These variations include the content of educational programs, professional recognition, and regulation [36,50]. 

The Australian Nursing and Midwifery Accreditation Council (ANMAC) is the accrediting body of higher education programs for nurses and midwives in Australia. To become a nurse or midwife in Australia, individuals must complete a program that has been accredited by the ANMAC and approved by the Nursing and Midwifery Board of Australia (NMBA). A review of the ANMAC midwife accreditation standards 2021 revealed a limited focus on neonatal education [51]. When used in conjunction with the NMBA midwifery standards for practice and the ICM essential competencies for midwifery practice, however, the minimum knowledge competencies regarding the care of the infant include those who are born low birth weight or preterm as well and care of the “special care baby”. Additionally, recognition of normal physiology as well as deviations from normal physiology are also essential competencies of the practicing midwife. Considering this, from an educational perspective, the care of babies with additional care needs could be possible in postnatal wards. de Rooy and Johns [34] described the various clinical considerations of what they termed the vulnerable infant, including late preterm infants, and found these babies could be cared for with and by their mothers in the postnatal ward. Their position was that these babies could be safely cared for using knowledge of an infant’s normal physiology but with due regard for evidence-based practice in preventing and responding to complications and problems should they arise [34]. 

There is an emerging theme in the literature that when trying to establish couplet care models for mothers and babies with additional care needs, facilities have needed to invest in extensive training and education for both nurses and midwives [3,11,17,29,30,34,52]. Similarly, other studies that implemented an educational intervention aimed at nurses and midwives in postnatal wards caring for late preterm infants found that education increased self-reported confidence and competence in caring for this patient population and their mothers [31,32]. Lunze et al. [27] found in their systematic review, similar themes with regards to education. They emphasised the importance of education for professional midwives to expand their scope and enhance their skills, knowledge, and competence in the care of vulnerable infants in the postnatal period [27]. Unsurprisingly, this finding was not isolated to midwives in the postnatal wards. A qualitative study undertaken in Scotland around the experiences of newly qualified paediatric nurses working in a neonatal unit highlighted that there were educational differences between nurses and midwives in the context of caring for babies and the postnatal mother in the neonatal nursery [30]. The study found distinct differences in the professional roles, knowledge, and inclination towards the care of the mother or baby between nurses and midwives. 

There are a few published studies of successful couplet care implementation that include the infant identified as having additional care needs [11,29]. It was pointed out, however, that those who implemented couplet care did not do so without extensive education and training of their nurses and midwives prior to introducing this model of care. For instance, when reorganising their healthcare team to be able to function in a newly built integrated mother–baby ward, Stelwagen et al. [29] described providing cross-training to members of the existing healthcare team so that most mother–baby couplets could be cared for in the integrated unit, including those with high and complex care needs [29]. The way in which they overcame the staffing challenge was through education and assigning patients with different levels of care to different nurses based on their prior experience and the units in which they worked before the integrated mother–baby care unit was established [29]. 

## 5. Discussion

Overall, this review revealed five main themes in relation to the nursing and midwifery barriers to couplet care of the mother and her infant who has additional care needs in postnatal wards. The themes revealed were safety, resistance, education, models of care and systems and other barriers. Resistance to couplet care was discussed as being caused by feelings of lack of confidence and competence [32], concerns around maternal and infant safety [5,43,44,45], and under-recognition of the benefits of couplet care [26]. 

While under-recognition of the importance of the postnatal period remains, so may the absence of adequate maternity models of care for this stage of the childbearing experience [35]. This review uncovered several articles that discussed alternative and innovative models of care such as transitional beds that have shown promising results. particularly in reducing separation in the postnatal environment [1,9,11,29,33,34,37]. 

However, education and training of existing nursing and midwifery staff were deemed necessary in all successfully implemented models found in the literature [36]. Education was the most prominent nursing and midwifery barrier cited in the literature with inconsistencies in the education of caring for vulnerable infants being the most evident both nationally and internationally [1,9,11,29,33,34,37]. Additionally, education featured as a barrier within some of those models of care, further highlighting how important education is in relation to postnatal care. This is not just about the education of midwives caring for these infants; it is also about the education of nurses caring for women who visit their infants in the nursery.

## 6. Limitations

While this review gained insight into the challenges of implementing couplet care internationally, it must be acknowledged that the existence of professional and educational differences was found among nurses and midwives worldwide. Definitions of infants requiring additional care also varied across the literature. These differences make it difficult to clearly identify the barriers to couplet care. In addition, couplet care is minimally being implemented and ascertaining the barriers is difficult to extract from this limited literature. This contributed to the small sample size and the variety of methodologies, further limiting this review. The small sample is another limitation, and maybe extending the search period may have helped but would have been impeded by changes in practice that naturally occur over time. 

This was not a systematic review, which has a more rigorous process of assessing the articles but still requires a review of research articles. Instead, this integrative review contained a variety of methodologies for researching as well as discussing couplet care, and not all research papers make the findings easy to apply, which adds a further limitation. There are a limited number of research articles that discuss the results of implementing couplet care and why an integrative review was adopted here. Despite the purposeful strategy used to identify the search terms and search for literature, it is possible that relevant articles were not identified, which may have been found with different search terms or databases. This could also account for the small sample size. Further, there is very limited literature available in the context of the baby requiring additional care in the postnatal ward in the couplet care model in Australia. The scope of this research being an Honours could also be considered a limitation.

## 7. Conclusions

Based on this review of the literature, it can be concluded that there is still a paucity of research in relation to barriers to couplet care. Since the conception of family-centred care and the subsequent evolution of mother–baby care, there have been minimal advances in the improvement of postnatal couplet care of the mother and her infant. This review intended find the barriers to couplet care of the mother and her infant with additional care needs; however, there was paucity of qualitative research of the experiences of nurses and midwives. One of the themes that strongly emerged from this review was the need for education of both nurses and midwives to competently provide couplet care, but this is not the only barrier identified here. Although this review discusses barriers to couplet care, more specific original research on what nurses and midwives themselves perceive to be the barriers to couplet care in Australia is needed. Further research on this cohort in the context of couplet care warrants investigation. Specifically interviewing nurses and midwives to ascertain what they perceive are the issues to implementing couplet care.

## Figures and Tables

**Figure 1 healthcare-11-00737-f001:**
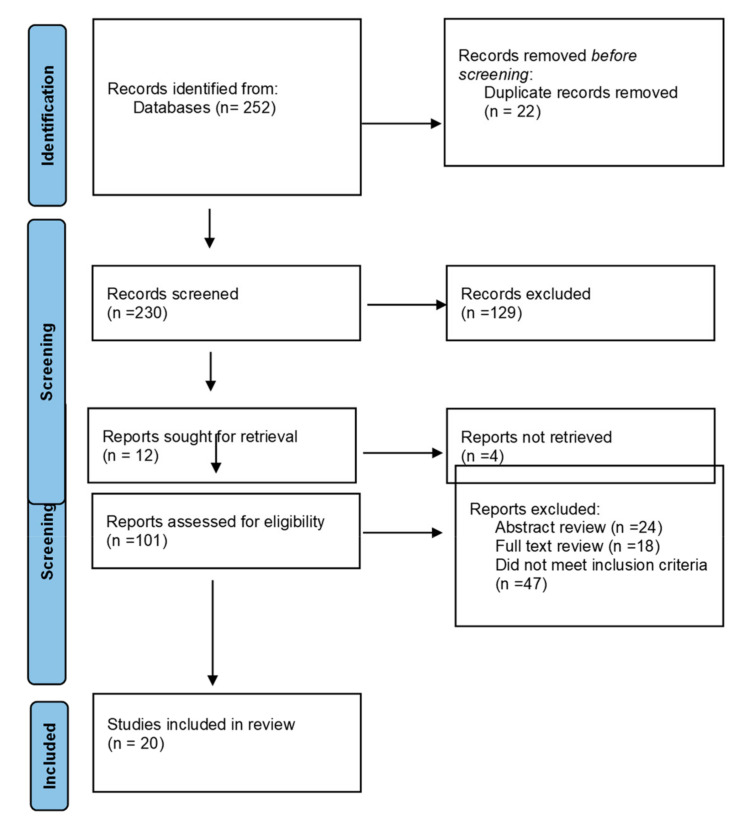
PRISMA flow diagram of search process.

**Table 1 healthcare-11-00737-t001:** PEO framework for search strategy.

Population and their problems	Nurses and midwives in postnatal wards caring for mothers and their babies who have additional care needs
Exposure	Couplet care
Outcome	Identify barriers

**Table 2 healthcare-11-00737-t002:** Database syntax search results.

Data Bases Searched	PubMed via OVID	CINHAL	Cochrane Database	Scopus	Medline via OVID
Syntax search terms	Nurs AND/OR Midwi	Nurs AND/OR Midwi	Nurs AND/OR Midwi	Nurs AND/OR Midwi	Nurs AND/OR Midwi
	Postnat OR Post-partum	Postnat OR Post-partum	Postnat OR Post-partum	Postnat OR Post-partum	Postnat OR Post-partum
	Barrier AND/OR Facilitators OR Enablers	Barrier AND/OR Facilitators OR Enablers	Barrier AND/OR Facilitators OR Enablers	Barrier AND/OR Facilitators OR Enablers	Barrier AND/OR Facilitators OR Enablers
	Couplet care OR Mother-baby care OR Rooming-in OR Mother-infant dyad	Couplet care OR Mother-baby care OR Rooming-in OR Mother-infant dyad	Couplet care OR Mother-baby care OR Room OR Mother-infant dyad	Couplet care OR Mother-baby care OR Rooming OR Mother-infant dyad	Couplet care OR Mother-baby care OR Rooming-in OR Mother-infant dyad
	Late preterm OR Preterm AND Bab OR Infant OR Neonat	Late preterm OR Preterm AND Bab OR Infant OR Neonat	Late preterm OR Preterm AND Bab OR Infant OR Neonat	Late preterm OR Preterm AND Bab OR Infant OR Neonat	Late preterm OR Preterm AND Bab OR Infant OR Neonat
	Limiters: English language, NOT neonatal intensive care	Limiters: English language, NOT neonatal intensive care	Limiters: English language, NOT neonatal intensive care	Limiters: English language, NOT neonatal intensive care	Limiters: English language, NOT neonatal intensive care
Results	123 results	26 results	19 results	68 results	16 results

## Data Availability

No new data was created with this article.

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
