# Peer review of "Barriers to Couplet Care of the Infant Requiring Additional Care: Integrative Review"

_healthcare, 2023, doi:10.3390/healthcare11050737_

Round 1

Reviewer 1 Report

Abstract:

Aim: Please write a complete statement that covers the population, setting, etc,.

Results

Results: This review revealed 15 five main themes: safety, resistance, education, models of care and systems, and other barriers. This is not a complete sentence, what five main themes are barriers for what??

Introduction

The introduction needs to focus on why this review is important and how this work would add to the value of existing literature. The introduction section did not cover this important aspect. Moreover, also discuss the objective of this review in the introduction section. 

Methods

The method section did not clearly explain the study setting. Either this review covers the global data or some specific geographic location.

Moreover, why did you select the PEO guide to conduct this review there are other various comprehensive guides to carry out the review...any specific reason? Mention the reference to the PEO guide. 

Table 02: search item column is empty. Mention which terms were used in detail. 

Major revisions

Result

I would suggest summarizing the 20 papers and their key findings according to themes in a table. The table is missing and the authors just presented a descriptive explanation. Individual studies' findings are important for the reviewers to understand the context of the study. 

Discussion

The discussion section is missing and maybe the authors discussed the along with the study findings. I would suggest writing the result section explicitly apart from the discussion section.

The strengths and limitations of this study are missing. 

Author Response

Thank you for the comments and help to strengthen this article. Please see attached file.

Reviewer 2 Report

Review of article: Barriers to couplet care of the infant requiring additional care: integrative review

 Thank you for the opportunity to review this article, which was generally well written and interesting. I believe this would be a valuable addition to the literature with some restructuring and added attention to detail. I have outlined my recommendations below:

·         There’s some inconsistency and confusion about the actual focus of the article. The search terms include specifically midwifery and nursing barriers to couplet care and on p3, line 105, this is stated to be the focus but the results discuss a range of barriers. This needs to be addressed in the introduction – that the literature is complex and while this was the initial focus, the literature review revealed barriers at a range of levels, not just from the midwifery/nursing angle.

·         What do you mean by an ‘integrative review’ in your title? This should be defined in the introduction.

·         The introduction should start with the definition of ‘couplet care’ (currently on p 2, line 71) and outline/summarise exactly what this article provides – this should include a definition of ‘integrative review’ and centre your background on what is known about couplet care rather than the history of western biomedical mother/infant care models work. At the moment your background neglects to mention indigenous models, which is a serious omission. However, I would encourage you to focus more on the known benefits and challenges of couplet care and then outline why you have undertaken this literature review.

·         Is there any research about the frequency/likelihood/proportion of couplet care? The only citation that is referenced is Lundeen noting that it isn’t ‘consistently provided’.

·         You should also decide whether you are taking an international or national perspective. This could be detailed in the introduction as a way to ground the work. Currently, in the section of the results that addresses Education (4.3) you provide detail on the Australian context, which has little relevance to an international audience, unless you have already centred your work in the Australian context – which you could do when you explain your reasons for undertaking this work. Alternatively, you could take an international perspective and give a generalised overview of midwifery education (even if this is just to acknowledge its variation and possibility of not being a transferrable qualification across settings).

·         There are some issues with the description of your methodology, which need addressed. On page 4, lines 132-6 notes that more than half the articles found were removed after review for relevance and another 24 didn’t meet the the inclusion criteria. Examples of relevance and detail of the exclusion criteria is required. This could be in the text or as a footnote to Figure 1.

·         It would be great to have a table of the 20 included articles with columns for the date, authors, year, title, location, methodology and a very brief summary (eg n of participants/particularly relevant findings). This could be an appendix.

·         The limitations are not limitations but results, which should be woven into your results and discussion. Limitations would be things like the small number of articles identified, a small range of locations, a limited number of methodologies etc. Because you haven’t included any of that detail, the reader doesn’t know the limitations. Normally the limitations would be before the conclusion.

·         It’s a good idea to identify the themes you found in the articles but the themes are not discrete. 4.2 Resistance seems to be strongly linked to (if not a result of) 4.3 Education – those who aren’t well trained/confident in neonatal care are reluctant to support/resist couplet care. There are also parts of 4.5 Systems and other barriers, which relate to education  (eg line 338) and safety (lines 333-337). Perhaps revisit your themes and ensure that they really are different reasons for a lack of support for couplet care. 4.4 models of care should be described first because there is reference to models of care in 4.1 safety and 4.2 resistance (which could be combined with education in my opinion).

·         At the moment the results include a range of discussion that would be better placed in a separate discussion section or in the introduction. The results should summarise only the key findings under each heading, synthesising the results across studies, noting how they support or contradict each other. You can also include any limitations or concerns that you have regarding any of the work. Then the discussion would link your findings with what is already known about why couplet care isn’t widely practiced. Much of the detail from the results could be summarised in a discussion section that is started with the important point that you found few articles that met your search criteria and then using the 4.6 section Summary of findings to structure your general discussion around.

·         The majority of the work is well written but there are a few sentences that need a rewrite for clarity. This is can be addressed in the restructure. NB typo on line 336 – issue’s.

Author Response

Thank you for your comments and suggestions to contribute to strengthening this article. Please see attached file for addressing these comments.

Round 2

Reviewer 1 Report

Thanks for sharing the revised draft. The authors have improved the draft. However, some components still need to be addressed again.

The headline of table 01 is "PEO" and the headline of table 02 is "search strategy". I would suggest writing a meaningful headline instead of writing a/some word/s.

The detailed syntax of each database (PubMed via OVID CINHAL Cochrane database Scopus Medline via OVID) as a supplementary file is still missing.

The authors have written a section of the discussion section. I would suggest authors review the discussion section of previously published systematic review papers to revise and complete this section.

Author Response

Reviewer 1:

Reviewer 1

heading on tables changed

this is an integrative review as such the authors have searched through the databases to find different integrative reviews undertaken in the area of nursing and midwifery and have followed what the majority of articles undertake for an integrative review. There are even a number of articles on how to do an integrative review that do not outline a formula or template. The diversity is immense making it difficult to do one way. Unlike a systematic review which has a formula to follow. The authors have done what appears to be the majority of integrative review articles follow.

corrections have been undertaken to strengthen these points.

Reviewer 2

have failed to address some of the major problems - notably clarifying that the background to the article is very Western focused with no acknowledgement of indigenous or pre-medical model practice

Acknowledged that this is a Western focused review and that prehospital model practiced has been identified in the background as care in the home. As mentioned previously Indigenous practice in Australia is a whole different scenario and not within the scope of this article.

doesn't address the known benefits and risks of couplet care.

These have been made more explicit

little synthesis of the review articles and some are discussed in detail in the results section.

synthesis of review articles added under results section

The discussion hasn't engaged with the existing literature 

have added the references in discussion

it's still not exactly clear what an 'integrative review' actually is (ie how does it differ from a systematic review?)

have added definition of an integrative review and outlined difference to systematic review in limitation. This is more than what is included in integrative reviews. A search of a number of integrative reviews in nursing was undertaken included articles that discussed how to d an integrative review - none of them included what this reviewer is asking to be included in this article and is not within the scope of this article.

but still doesn't indicate where/how so many articles were excluded at
different stages in the review process. 

this has been outlined under screening

The limitations have been moved and slightly edited but are still not the limitations of the research process 

more limitations have been added as suggested plus review of other integrative reviews to assess what others have included

The authors thank the reviewers for their comments in making this a stronger paper.

Reviewer 2 Report

I recommended major changes and their response has mostly involved some reordering and minor clarification. The addition of the table of articles reviewed is excellent but they have failed to address some of the major problems - notably clarifying that the background to the article is very Western focused with no acknowledgement of indigenous or pre-medical model practice and doesn't address the known benefits and risks of couplet care. They also have not addressed the issues around the results where there is little synthesis of the review articles and some are discussed in detail in the results section. The discussion hasn't engaged with the existing literature and is still in its existing form. The edits to the article appear rushed and haven't really engaged with the review issues - ie it's still not exactly clear what an 'integrative review' actually is (ie how does it differ from a systematic review?) and the exclusion criteria was moved in the article but still doesn't indicate where/how so many articles were excluded at different stages in the review process. The additions feel rather rushed
- eg in order to justify leaving the background largely the same, there is a note on P2 that the literature is mostly international but there's a bit of an Australian focus because this was an Australian research project. The limitations have been moved and slightly edited but are still not the limitations of the research process - these would be things like issues with search terms, change in practice over time, the relatively small number of articles found, challenges in the research process. 

Author Response

Reviewer 2

have failed to address some of the major problems - notably clarifying that the background to the article is very Western focused with no acknowledgement of indigenous or pre-medical model practice

Acknowledged that this is a Western focused review and that prehospital model practiced has been identified in the background as care in the home. As mentioned previously Indigenous practice in Australia is a whole different scenario and not within the scope of this article.

doesn't address the known benefits and risks of couplet care.

These have been made more explicit

little synthesis of the review articles and some are discussed in detail in the results section.

synthesis of review articles added under results section

The discussion hasn't engaged with the existing literature

have added the references in discussion

it's still not exactly clear what an 'integrative review' actually is (ie how does it differ from a systematic review?)

have added definition of an integrative review and outlined difference to systematic review in limitation. This is more than what is included in integrative reviews. A search of a number of integrative reviews in nursing was undertaken included articles that discussed how to d an integrative review - none of them included what this reviewer is asking to be included in this article and is not within the scope of this article.

but still doesn't indicate where/how so many articles were excluded at
different stages in the review process.

this has been outlined under screening

The limitations have been moved and slightly edited but are still not the limitations of the research process

more limitations have been added as suggested plus review of other integrative reviews to assess what others have included

The authors thank the reviewers for their comments in making this a stronger paper.